# Causal evidence for lateral prefrontal cortex dynamics supporting cognitive control

**Derek Evan Nee[1]\*, Mark D'Esposito[2,3]**

[1]Department of Psychology, Florida State University, Tallahassee, United States; [2]Helen Wills Neuroscience Institute, University of California, Berkeley, Berkeley, United States; [3]Department of Psychology, University of California, Berkeley, Berkeley, United States

**Abstract** The lateral prefrontal cortex (LPFC) is essential for higher-level cognition, but the nature of its interactions in supporting cognitive control remains elusive. Previously (Nee and D'Esposito, 2016), dynamic causal modeling (DCM) indicated that mid LPFC integrates abstract, rostral and concrete, caudal influences to inform context-appropriate action. Here, we use continuous theta-burst transcranial magnetic stimulation (cTBS) to test this model causally. cTBS was applied to three LPFC sites and a control site in counterbalanced sessions. Behavioral modulations resulting from cTBS were largely predicted by information flow within the previously estimated DCM. However, cTBS to caudal LPFC unexpectedly impaired processes that are presumed to involve rostral LPFC. Adding a pathway from caudal to mid-rostral LPFC significantly improved the model fit and accounted for the observed behavioral findings. These data provide causal evidence for LPFC dynamics supporting cognitive control and demonstrate the utility of combining DCM with causal manipulations to test and refine models of cognition.
DOI: https://doi.org/10.7554/eLife.28040.001

**\*For correspondence:**
derek.evan.nee@gmail.com

**Competing interests:** The authors declare that no competing interests exist.

## Introduction

Context-appropriate behavior requires assessment of both present circumstances and future plans to determine the best course of action in the moment. Guiding behavior in accordance with internal representations rather than habitual stimulus-response tendencies requires cognitive control that depends on the lateral prefrontal cortex (LPFC; *Miller and Cohen, 2001*). Yet, how the functional properties and interactions among areas of the LPFC support cognitive control remains poorly understood.

Previously, we collected fMRI data on human participants to investigate how the LPFC supports cognitive control by examining intercommunication among LPFC areas across a variety of cognitive control demands (*Figure 1A,B*; *Nee and D'Esposito, 2016*). First, we used univariate analysis to determine the functional responses of different LPFC areas. This analysis revealed that caudal LPFC was responsive to attention to stimulus features (*Feature Control*), mid LPFC was responsive to contextual rules to be applied to the attended stimulus feature (*Contextual Control*), and rostral LPFC was responsive to retaining a past stimulus feature for a future rule (*Temporal Control*). Collectively, these results supported the hypothesis that progressively rostral LPFC areas perform progressively abstract processes (*Fuster, 2001*; *Koechlin et al., 2003*; *Badre, 2008*; *Badre and D'Esposito, 2009*). Second, we used dynamic causal modeling (DCM) to examine how intercommunication among LPFC areas supports cognitive control (*Friston et al., 2003*). The estimated model (*Figure 1D*) demonstrated three principle properties: (1) caudal LPFC provides feature-specific inputs to the rest of the LPFC; (2) mid-rostral and caudal influences converge in mid LPFC during

*Contextual Control*; and (3) the rostral-most LPFC remains largely segregated from the rest of the LPFC during *Temporal Control* (i.e. functional modulations do not extend to the rest of LPFC; *Figure 1D*, red arrows). Collectively, these results indicated that the mid LPFC is a nexus where multiple influences converge to guide context-appropriate action, providing a new framework upon which to understand how the functional interactions within the LPFC support cognitive control.

A central goal of neuroscience is to determine causal associations among stimuli, neural systems, and behavior as a gateway to specifying predictive models of behavior. Although DCM has been demonstrated to detail functional neural circuitry accurately (*Lee et al., 2006*; *David et al., 2008*; *Bernal-Casas et al., 2017*) modeling causal links among neural regions, it does not specify the causal relationship between neural interactions and behavior. Given the complexities of neural interactions, however, a model of directed influences is a critical intermediary to determine the link between brain and behavior (*Jazayeri and Afraz, 2017*). That is, a model of directed influences affords predictions regarding how perturbations of specific brain regions affect other regions and ultimately behavior. Within this framework, one can identify parent and children nodes wherein perturbations of parents affect children, but not vice versa. This sort of framework is critical to establish neural hierarchies (*Badre et al., 2009*; *Azuar et al., 2014*), and to navigate the complex pathways by which neural activity leads to behavior.

Here, we apply this logic by using continuous theta-burst transcranial magnetic stimulation (cTBS) to reduce cortical excitability reversibly (*Huang et al., 2005*). We apply cTBS to either caudal (superior frontal sulcus; SFS), mid (ventrolateral prefrontal cortex; VLPFC), or rostral (lateral frontal pole; FPl) LPFC, as well as to a control site (primary somatosensory cortex; S1) in a within-subjects counterbalanced design (*Figure 1C*). Following the previously estimated DCM, we predicted that (1) cTBS to caudal LPFC would result in a *feature-specific* impairment acting generally across cognitive control demands. This follows from the model prediction that caudal LPFC provides feature inputs to the rest of the LPFC system. (2) cTBS to mid LPFC would result in a *feature-specific* impairment during *Contextual Control*. This follows from the model prediction that mid LPFC integrates feature information from caudal LPFC and task information from mid-rostral LPFC to perform *Contextual Control*. (3) cTBS to the rostral-most LPFC would result in a *feature-general* impairment during *Temporal Control*. This follows from the model prediction that the rostral-most LPFC is functionally segregated from other LPFC areas during *Temporal Control* and that this region is insensitive to feature information. These patterns of results would provide causal evidence linking the estimated neural dynamics to their presumed behavioral correlates.

## Results

### Replication of previous fMRI results

Prior to receiving cTBS, all participants underwent an fMRI session using the task described previously (*Figure 1A,B*; *Nee and D'Esposito, 2016*) in order to localize individual targets for cTBS (*Figure 1C*). The new sample offered an opportunity to replicate the previous findings. Each individual performed a single fMRI session in the present study compared to two sessions in the previous work, so the effects reported here are expected to have reduced power relative to those in the original study. Nevertheless, as depicted in *Figure 2* and its supplements, virtually all of the previously reported effects were replicated (replication statistics reported in the figure captions). As before, a progression of activation from caudal to rostral LPFC was observed for *Feature Control* to *Contextual Control* to *Temporal Control* (*Figure 2*). Sensitivity to stimulus features was present in caudal and mid, but not rostral LPFC consistent with more abstract processing in rostral LPFC (*Figure 2— figure supplement 1*). Caudal LPFC was related to present but not future behavior, rostral LPFC showed the inverse pattern, and mid LPFC showed sensitivity to both present and future behavior, collectively forming a temporal abstraction gradient (*Figure 2—figure supplement 2*). Finally, the effective connectivity parameters estimated by DCM were similar to those in the previous report. The major difference from the previously reported DCM results was that the pathways linking caudal to mid LPFC were not modulated by attention to stimulus features (i.e. *Stimulus Domain*). Instead, the stimulus inputs to caudal LPFC were increased several fold (*Figure 2—figure supplement 3*).

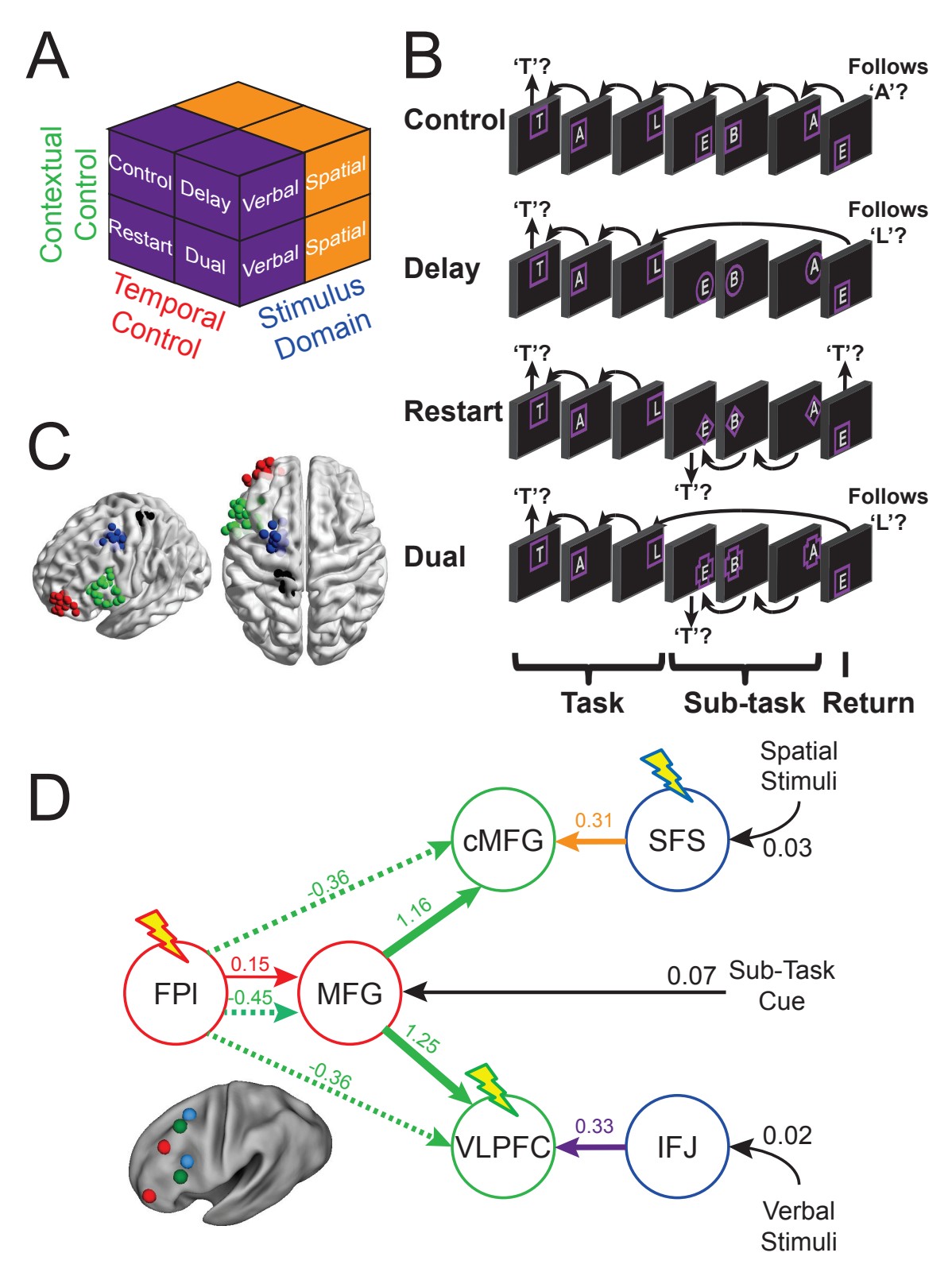

**Figure 1.** Task, cTBS targets, and dynamic causal model. (**A**) The design orthogonally manipulated factors of *Stimulus Domain* (verbal, spatial) and two forms of cognitive control: *Contextual Control* (low – *Control, Delay*; high – *Restart, Dual*) and *Temporal Control* (low – *Control, Restart*; high – *Delay, Dual*). These factors were fully crossed in a 2 × 2 × 2 design. (**B**) The basic task required participants to judge whether a stimulus followed the previous stimulus in a sequence. The sequence in the verbal task was the order of the letters in the word 'tablet.' The sequence in the spatial task was a trace of

*Figure 1 continued on next page*

*Figure 1 continued*

the points of a star. The start of each sequence began with a decision regarding whether the currently viewed stimulus is the start of the sequence (e.g. 't' in the verbal task). Factors were blocked with each block containing a basic task phase, a sub-task phase, a return trial, and a second basic task phase (not depicted), for all but the *Control* blocks. *Control* blocks consisted only of the basic task phase extended to match the other conditions in duration. Colored frames indicated whether letters or locations were relevant for the block (verbal – purple; spatial – orange in this example; verbal condition depicted). The basic task was cued by square frames. Other frames cued the different sub-task conditions. In the *Delay* condition (circle frames), participants held in mind the place in the sequence across a distractor-filled delay. In the *Restart* condition (diamond frames), participants started a new sequence. In the *Dual* condition (cross frames), participants simultaneously started a new sequence, and maintained the place in the previous sequence. (C) Each sphere represents a stimulus target for an individual. Red – rostral LPFC, green – mid LPFC, blue – caudal LPFC, black – S1 (control site). (D) The dynamic causal model estimated previously (*Nee and D'Esposito, 2016*). Colored arrows denote modulations of effective connectivity during different cognitive control demands (orange – attention to spatial features; purple – attention to verbal features; green – *Contextual Control*; red – *Temporal Control*). Colors of the circles denote univariate sensitivities to *Feature Control* (blue), *Contextual Control* (green), and *Temporal Control* (red). Lightning bolts indicate targets for continuous theta-burst transcranial magnetic stimulation (cTBS). cTBS was predicted to affect behavior for which a given region was responsive, and also behaviors supported by downstream regions that require processing in upstream targets. cMFG, caudal middle frontal gyrus; FPl, lateral frontal pole; IFJ, inferior frontal junction; MFG, middle frontal gyrus; SFS, superior frontal sulcus; VLPFC, ventrolateral prefrontal cortex.

DOI: https://doi.org/10.7554/eLife.28040.002

Intuitively, these different findings reflect whether an attention-related gain is realized prior to (input parameter) or after (modulation of caudal to mid LPFC effective connectivity) feature information arrives at LPFC. Effectively, this leads to the same result – feature information is propagated through the LPFC via input nodes in feature-specific caudal LPFC during attention to a given feature. Statistically, these different findings reflect an attentional gain that is sustained (modulation) versus transient (input). These differences do not affect the predictions of the model with regard to cTBS because either way, disruption of caudal LPFC is predicted to disrupt feature-specific inputs to the rest of the LPFC.

As before, additional analyses on the effective connectivity parameters estimated by DCM were performed to examine hierarchical control and its relationship to higher-level cognitive ability, as indexed by a combination of short-term/working memory capacity and fluid intelligence (see *Nee and D'Esposito, 2016*) for complete details). Replicating the previous results, hierarchical strength, as measured by greater efferent relative to afferent fixed connectivity, peaked in mid-rostral LPFC and was low in FPl, the rostral-most area, suggesting that the rostral-most LPFC is not the apex of the hierarchy (*Figure 2—figure supplement 4*). Once again, individuals varied in the strength of bottom-up versus top-down modulations as a function of cognitive control with a trade-off between the two reflected in a negative correlation between these parameters (*Figure 2—figure supplement 5A*). However, whereas we had previously reported that both individual differences in top-down modulations as a function of cognitive control (*Figure 2—figure supplement 5B*) and hierarchical strength in fixed connectivity (*Figure 2—figure supplement 5D*) were positively related to individual differences in trait-measured higher-level cognitive ability, this relationship was not evident in the present sample. A null effect might be expected given the reduced power in the present sample relative to the previous one, but the trends that were evident were opposite in sign to those observed previously. As reported in more detail below, the positive relationship between DCM parameters and trait-measured higher-level cognitive ability were also not replicated in the revised DCM. Given that nearly all of the other effects were replicated (over 50 independent tests performed), this particular lack of replication may indicate that our previous result was a false positive.

## Behavioral results and effects of cTBS

To examine the replication of the previously reported behavioral effects of cognitive control demands, separate 2 × 2 × 2 ANOVAs on error-rate (ER), and reaction times (RT) on correct trials were performed for data collected during the fMRI session during the sub-task phase. As before, these analyses revealed significant effects of cognitive control demands with main effects of *Temporal Control* (ER – $F(1,23) = 6.04$, $p<0.05$; RT – $F(1,23) = 11.34$, $p<0.005$), *Contextual Control* (ER – $F(1,23) = 60.30$, $p<10^{-7}$; RT – $F(1,23) = 149.48$, $p<10^{-10}$), and their interaction (ER – $F(1,23) = 6.92$, $p<0.05$; RT – $F(1,23) = 105.30$, $p<10^{-9}$). While no main effect of *Stimulus Domain* was observed previously, in the present sample there was a main effect of *Stimulus Domain* in RT ($F(1,23) = 4.28$,

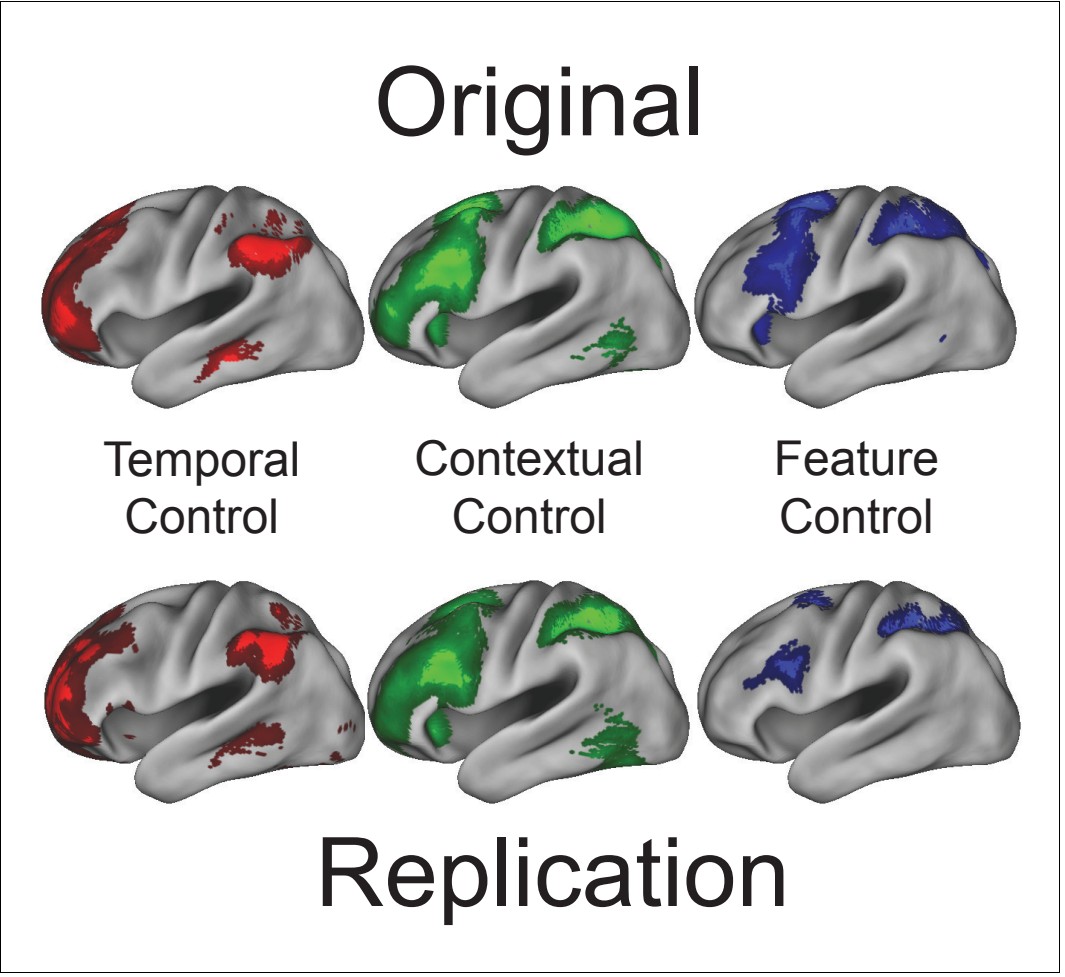

**Figure 2.** Univariate replication.  Top: previously reported univariate effects. Bottom: univariate effects in the present study. In both cases, *Temporal Control* elicited activation of rostral LPFC, *Contextual Control* of mid LPFC, and *Feature Control* of caudal LPFC, indicating a rostral-caudal gradient of cognitive control. All results thresholded at p<0.001 at the voxel level with cluster extent providing family-wise error correction at p<0.05.
DOI: https://doi.org/10.7554/eLife.28040.003

The following figure supplements are available for figure 2:

**Figure supplement 1.** Stimulus domain abstraction replication.
DOI: https://doi.org/10.7554/eLife.28040.004

**Figure supplement 2.** Temporal activation-behavior relationship replication.
DOI: https://doi.org/10.7554/eLife.28040.005

**Figure supplement 3.** LPFC dynamic causal model replication.
DOI: https://doi.org/10.7554/eLife.28040.006

**Figure supplement 4.** Hierarchical fixed dependencies replication.
DOI: https://doi.org/10.7554/eLife.28040.007

**Figure supplement 5.** LPFC dynamics and higher-level cognitive ability replication failure.
DOI: https://doi.org/10.7554/eLife.28040.008

p<0.05) but not ER (F(1,23) = 2.33, p>0.1). Participants performed more quickly on the spatial task relative to the verbal task. Performance on the spatial relative to verbal task improved with practice as is evidenced by an increase in the effect size during the cTBS sessions as reported below. Once again, there were no interactions between *Stimulus Domain* and cognitive control demands (ER and RT all p>0.25), although this also changed with practice as detailed below.

Next, we examined effects of cTBS on behavior. cTBS targets were determined on the basis of individual activation peaks (see Materials and methods for full details). To test predictions

that were based upon the previously estimated DCM (*Nee and D'Esposito, 2016*), one target was chosen from each of caudal, mid, and rostral LPFC. Of the caudal areas, SFS was the most accessible to stimulation. VLPFC in mid LPFC, and FPl in rostral PFC were subsequently chosen as targets that maximized between-target distance, which was anticipated to most readily yield dissociable patterns of stimulation. The full uncontrasted data are depicted in *Figure 3A–B*. Given that the hypotheses are borne out of interactions that can be difficult to visualize, we depict contrasts that address model predictions in *Figure 3C–F*. For each contrast of interest, linear mixed effects models were estimated including factors of *Target* (FPl, VLPFC, SFS, S1), *Time* (i.e. session number), and their interaction. Each model included random effects terms for *Subject*, and *Time* nested within *Subject* (i.e. individual differences in practice-related changes). As alluded to above, *Time* was included as a factor after changes in contrasts of interest were noted over sessions. Hereafter, we focus on the effects of *Target* as the factor of main interest. We also describe simpler ANOVAs in *Figure 3— figure supplement 1*.

First, the model predicts that cTBS to caudal LPFC would result in a feature-specific impairment given that caudal LPFC provides feature inputs to the rest of the LPFC. In this case, we targeted the caudal superior frontal sulcus (SFS), roughly corresponding to the frontal eye-fields, which was

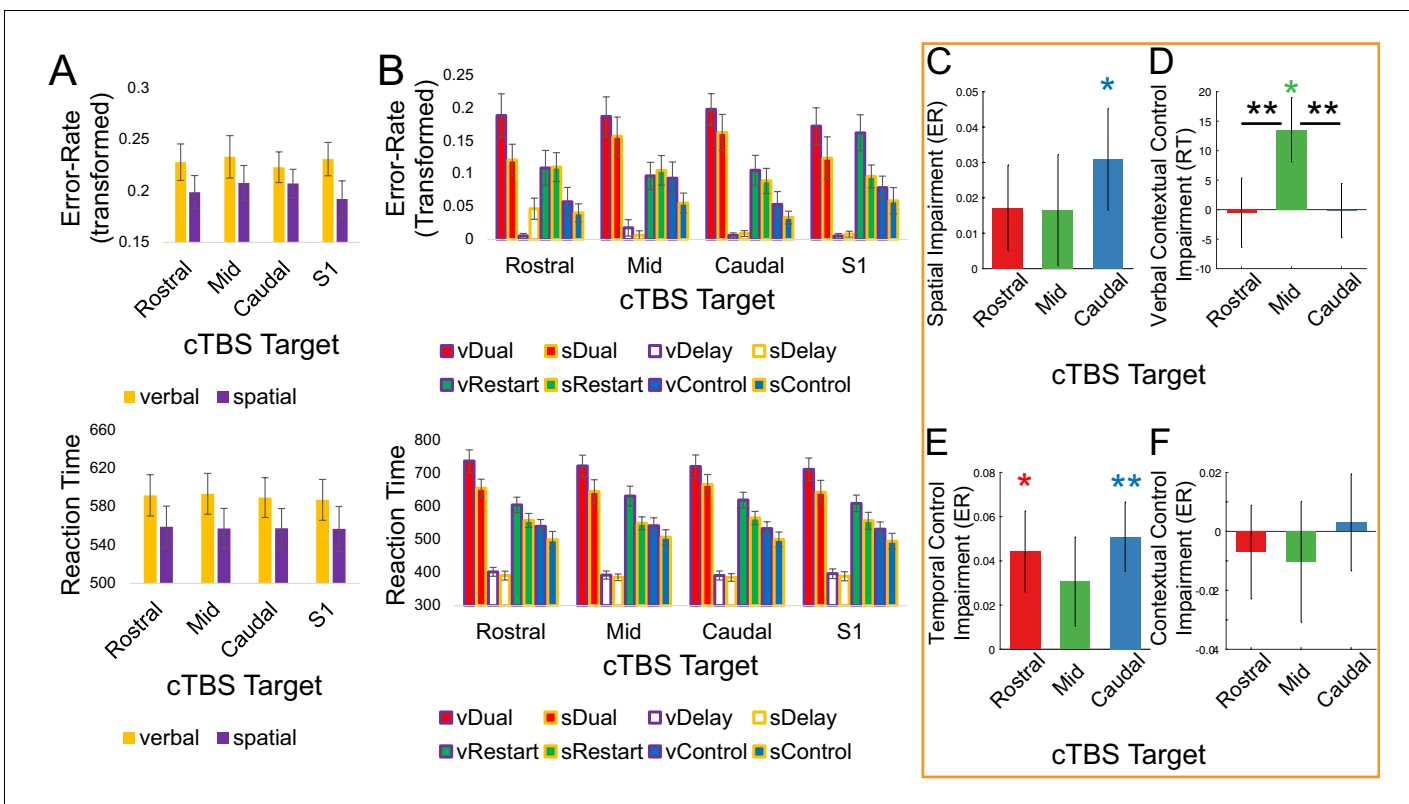

**Figure 3.** Behavioral effects following cTBS. Left: Uncontrasted data from (**A**) all trials averaged across the verbal and spatial tasks, respectively, and (**B**) sub-task trials averaged across the eight factors of the task design. Right: Contrasted data. In all bars, performance following cTBS to the control target (S1) has been subtracted out such that the y-axis indicates a behavioral impairment relative to control stimulation. Effects of *Time* have been regressed out (see *Figure 3—figure supplement 1* for data that do not control for effects of *Time*). The x-axis indicates the stimulation target (rostral – rostral LPFC/FPl; mid – mid LPFC/VLPFC; caudal – caudal LPFC/SFS). (**C**) Impairments in *Spatial Stimulus Domain* relative to *Verbal Stimulus Domain* in error-rate (ER). (**D**) The interaction between *Stimulus Domain* x *Contextual Control* in reaction time (RT) contrasted such that relative verbal impairments are positive (i.e. greater verbal *Contextual Control* cost relative to spatial *Contextual Control* cost). (**E**) Impairments in the main effect of *Temporal Control* in ER. (**F**) Impairments in the main effect of *Contextual Control* in ER. *p<0.05; **p<0.005.

DOI: https://doi.org/10.7554/eLife.28040.009

The following figure supplement is available for figure 3:

**Figure supplement 1.** Contrasts without controlling for effects of *Time*.

DOI: https://doi.org/10.7554/eLife.28040.010

modeled as the source of spatial feature inputs. We expected diminished spatial relative to verbal task performance to result from cTBS to SFS. To test this prediction, we examined how the effect of *Stimulus Domain* was modulated by cTBS *Target*. Linear mixed effects models were performed on the effect of *Stimulus Domain* (contrast of verbal and spatial performance). This analysis revealed a significant effect of *Target* in ER (F(3,23.17) = 4.46, p<0.05), but not RT (F(3,37.98) = 1.363, p>0.25). Follow-up tests comparing cTBS to SFS and S1 after controlling for *Time* revealed that cTBS to SFS resulted in significantly reduced spatial relative to verbal task performance in ER (t(22) = 2.15, p<0.05; *Figure 3C*). While participants generally performed better on the spatial relative to verbal task after cTBS (FPl – t(22) = 3.67, p<0.005; VLPFC – t(22) = 2.43, p<0.05; S1 – t(22) = 3.04, p<0.01), this was not the case following cTBS to SFS (t(22) = 1.41, p>0.15; ). Hence, cTBS to SFS resulted in a specific impairment in spatial task performance in line with model predictions.

Second, the model predicts that cTBS to mid LPFC would result in a feature-specific impairment in *Contextual Control* given that *Contextual Control* requires integration of bottom-up feature information (from caudal LPFC) and top-down task information (from mid-rostral LPFC). In this case, we targeted the ventrolateral prefrontal cortex (VLPFC), which was sensitive to *Contextual Control* for verbal information. Correspondingly, we expected diminished verbal relative to spatial *Contextual Control* performance to result from cTBS to VLPFC. To test this prediction, we examined whether the *Stimulus Domain* x *Contextual Control* interaction was modulated by cTBS *Target*. Linear mixed effects models revealed a significant effect in RT (F(3,24.14) = 6.97, p<0.005), but not ER (F (3,39.73) = 1.28, p>0.25). Follow-up tests comparing cTBS to VLPFC and S1 after controlling for *Time* revealed significantly increased RT costs associated with *Contextual Control* for the verbal relative to spatial task following cTBS to VLPFC (t(21) = 2.43, p<0.05; *Figure 3D*). Similar differences were observed when comparing cTBS following VLPFC to FPl (t(21) = 3.19, p<0.005) and SFS (t(21) = 3.31, p<0.005). Thus, cTBS to VLPFC resulted in a feature-specific, control-specific impairment consistent with predictions of the model.

Finally, we examined the impact of cTBS on the rostral LPFC by targeting FPl. According to the model, effective connectivity from the FPl to mid LPFC is reduced during *Contextual Control*. In the previous report, we suggested that this negative modulation effectively nullifies positive associations between the FPl and mid LPFC in fixed connectivity, thereby serving to segregate the FPl from the rest of the LPFC. If so, cTBS to FPl would not be expected to have an effect on *Contextual Control* since cTBS is expected to perform a similar segregating function by way of reducing FPl cortical excitability. Another possibility is that the negative modulation reflects top-down inhibition from FPl to mid LPFC, which may serve to prioritize which contextual information is appropriate at which time. In this case, cTBS to FPl would be expected to affect *Contextual Control* since reducing FPl cortical excitability would, in turn, reduce the top-down bias it transmits. Either way, an effect on *Temporal Control* was anticipated given the region's sensitivity to *Temporal Control*. To test these predictions, we estimated linear mixed effects models separately for the effects of *Contextual Control* and *Temporal Control*. These analyses revealed no effect of *Target* on *Contextual Control* (ER – F(3,23.22) = 2.52, p>0.05; RT – F(3,27.82) = 0.43, p>0.7), but a significant effect of *Target* on *Temporal Control* in both ER (F(3,47.43) = 3.51, p<0.05) and RT (F(3,35.79) = 3.02, p<0.05). Follow-up tests comparing cTBS to FPl and S1 after controlling for *Time* revealed that FPl stimulation led to significantly increased *Temporal Control* costs in ER (t(21) = 2.41, p<0.05), but not RT (t(21) = 0.57, p>0.55). No such differences were observed on *Contextual Control* costs, which non-significantly trended in the opposite direction (i.e. reduced costs; ER – t(21) = −0.44, p>0.65; RT – t(21) = −1.05, p>0.3). Hence, these data indicate that the FPl is causally related to *Temporal Control* (*Figure 3E*), but do not provide evidence that cTBS to FPl affects *Contextual Control* (*Figure 3F*).

Collectively, the analyses above suggest that caudal, mid, and rostral LPFC support separable cognitive control processes, the nature of which is predicted by a model of LPFC dynamics. To further corroborate these associations, we performed a 4 × 3 ANOVA with factors of *Target* and *Contrast* using the three contrasts reported above that were predicted to demonstrate dissociative effects (*Stimulus Domain*, *Stimulus Domain* x *Contextual Control*, *Temporal Control*). This analysis confirmed a significant interaction (F(6,126) = 4.51, p<0.0005). A similar analysis extended to include the *Contextual Control* contrast was also significant (F(9,189) = 2.92, p<0.005). Although we cannot conclude a triple dissociation because not all pairwise differences were significant for each measure, this result adds credence to the association of each region to its corresponding model-predicted effect.

Upon depicting the results of the analysis above, an unanticipated finding emerged: cTBS to SFS also had a negative effect on *Temporal Control*. To detail this effect further, we directly compared cTBS to SFS and S1 after controlling for *Time*. Similar to the pattern observed with cTBS to FPl, cTBS to SFS significantly increased *Temporal Control* costs in ER (t(21) = 3.24, p<0.005), but not RT (t(21) = 0.41, p>0.65), but had no effect on *Contextual Control* costs (ER – t(21) = 0.18, p>0.85; RT – t(21) = 0.37, p>0.7). This result was not predicted by the model and no such effects were observed comparing cTBS to VLPFC with S1 (all tests p>0.1). However, it is important to note that using DCM requires two steps: first is a model comparison step wherein plausible models of neural dynamics are compared; and second are inferences on the parameters of the best model of the data. Given that SFS did not exhibit a positive univariate effect of *Temporal Control* in our original study, we did not include SFS modulations as a function of *Temporal Control* in our original model. However, taking the previously estimated model as a starting point, the SFS could potentially influence *Temporal Control* through interactions with the mid-rostral LPFC (MFG) or FPl. Thus, a behavioral impairment on *Temporal Control* following cTBS to SFS could be observed if SFS positively modulates MFG or FPl to facilitate *Temporal Control*.

## Network dynamics revisited

To examine whether the observed impact of SFS on *Temporal Control* could be due to a previously unappreciated modulation of SFS on rostral LPFC, we initially compared the model that we described previously (*Figure 1*; *Figure 2—figure supplement 3*) to models in which the SFS modulates processing in either the MFG or FPl during *Temporal Control*. This initial comparison revealed that models containing SFS-MFG connectivity and modulations were a substantially better match to the data (family model exceedance probability 0.9992). We then proceeded to explore the model space thoroughly assuming fixed connectivity between the SFS and MFG, and varying the pathways by which modulations of connectivity supported cognitive control.

Group-level Bayesian model comparison adjudicated between models of effective connectivity within the LPFC (*Stephan et al., 2009*) (see Materials and methods for full details of the procedure). Significant parameter estimates resulting from the revised best model (hereafter, the revised model) are depicted in *Figure 4* for the previously described data (*Figure 4—figure supplement 1A*), the present data (*Figure 4—figure supplement 1B*), and the data considered jointly (*Figure 4*). The major difference from the original model is that the feedback modulation from FPl to MFG as a function of *Temporal Control* has been replaced by a feedforward modulation from SFS to MFG. There is also evidence of a correspondingly negative feedback modulation from MFG to SFS. In this model, disruption of SFS by cTBS would be predicted to impair *Temporal Control* by disrupting feedforward influences from SFS to MFG. This modulation was not necessitated by the model comparison or estimation procedures. Models containing no feedforward modulation from SFS to MFG were directly compared to models containing this influence in several steps of the model space exploration. Furthermore, the model comparison procedure only posits what modulations might exist, not the sign or strength of the modulation. Hence, even within models containing the feedforward influence from SFS to MFG, the modulation could have been negative rather than positive (or even non-significant). That the best model of the data contained this positive modulation from SFS to MFG is therefore in striking agreement with the behavioral effects of cTBS, demonstrating the utility of combining models of regional dynamics with neural perturbations (*Jazayeri and Afraz, 2017*).

While the analyses above indicate that including a bidirectional modulation between SFS and MFG during *Temporal Control* provides the best fit of the data among the models tested, it remains possible that there may yet be other unexamined modulations that provide an even better fit. To examine the specificity of the proposed revised model, we explored whether adding bidirectional modulations during *Temporal Control* between each of the 4 other regions (IFJ, cMFG, VLPFC, and FPl) and MFG might further improve the model fit. These modulations were estimated in addition to, or in place of, the bidirectional modulation between SFS and MFG during *Temporal Control* for a total of eight additional models. These eight models were directly compared to the revised model described above using group-level Bayesian model comparison. In both the present (model exceedance probability = 0.52) and previous sample (model exceedance probability = 0.81), the revised model was clearly superior to these other potential models. Such data indicate that there is specificity to the interaction between SFS and MFG during *Temporal Control* and that simply adding additional modulations does not improve the model fit.

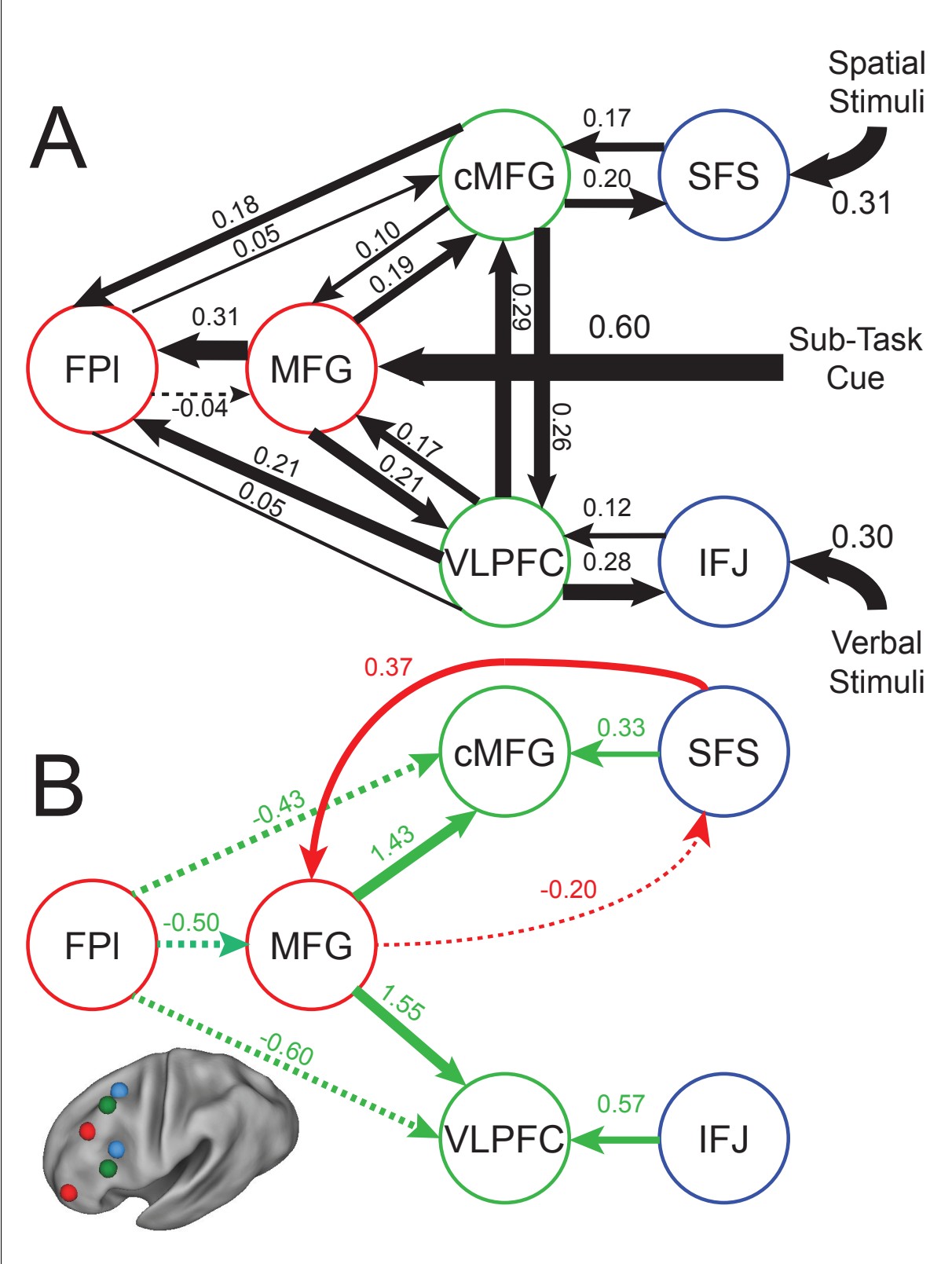

**Figure 4.** Revised LPFC dynamic causal model. Bayesian model selection indicated that the depicted model was the best model of the dynamics among the models tested. Arrows indicate direction of influence, numbers and line widths indicate the strength of influence, and dashed arrows indicated inhibitory influences. Parameter estimates have been averaged across the previous and present samples. (A) Fixed connectivity and inputs depicted in black. (B) Modulations of connectivity by *Contextual Control* (green), and *Temporal Control* (red) demands depicted in colors. All depicted

*Figure 4 continued on next page*

*Figure 4 continued*

parameters are significant after correction using false-discovery rate. cMFG, caudal middle frontal gyrus; FPl, lateral frontal pole; IFJ, inferior frontal junction; MFG, middle frontal gyrus; SFS, superior frontal sulcus; VLPFC, ventrolateral prefrontal cortex.

DOI: https://doi.org/10.7554/eLife.28040.011

The following figure supplements are available for figure 4:

**Figure supplement 1.** Revised LPFC dynamic causal model.

DOI: https://doi.org/10.7554/eLife.28040.012

**Figure supplement 2.** Revised hierarchical fixed dependencies.

DOI: https://doi.org/10.7554/eLife.28040.013

Finally, as in our original study, we examined measures of hierarchical strength and the relationship between model parameters and trait-measured higher-level cognitive ability. In the new model, effects of hierarchical strength remained highest in mid-rostral LPFC (both samples combined, positive peak of fitted vertex $t(47) = 11.63$, $p<10^{-14}$; mean position of vertex y = 21.7 in MNI space; *Figure 4—figure supplement 2*). However, no relationship was observed between model parameters and trait-measured higher-level cognitive ability (both samples combined, hierarchical strength – $r = -0.12$, $p>0.4$; top-down strength – $r = 0.18$, $p>0.2$; bottom-up strength $r = 0.10$, $p>0.4$). While these data cast doubts on whether the model parameters can predict trait-measured cognitive ability, the good agreement between the effects of cTBS on behavior and the model predictions add validity to the model's relationship to behavioral manifestations of cognitive control.

## Discussion

We used cTBS to test the causal relationship between a neural model of LPFC dynamics supporting cognitive control and behavioral manifestations of these control processes. In line with the model's predictions, cTBS to caudal LPFC disrupted stimulus feature processing, cTBS to mid LPFC disrupted the response to a stimulus feature based on prevailing contextual contingencies (*Contextual Control*), and cTBS to rostral-most LPFC disrupted the ability to balance ongoing and future demands (*Temporal Control*). Unexpectedly, cTBS to caudal LPFC also disrupted *Temporal Control*. This result could not be accommodated by our previously modeled connectivity dynamics (*Nee and D'Esposito, 2016*). However, a revised model that included a modulation from caudal LPFC to mid-rostral LPFC provided a better fit to the neural data and a ready explanation for the observed impairment. Collectively, these results demonstrate the mutually informative nature of models of neural dynamics and causal brain-behavior tests, and reveal how the LPFC supports cognitive control.

cTBS is presumed to reduce cortical excitability (*Huang et al., 2005*), providing the ability to modulate behaviors for which neural activity of a stimulated region is critical. It is commonly assumed that applying cTBS to regions that show increased activity during a particular behavior will disrupt that behavior insofar as that area is necessary. With this logic in mind, it may be surprising to find that cTBS to caudal LPFC impacted *Temporal Control* as this area showed *decreased* rather than increased activity during *Temporal Control* (previous/present sample: $F(1,23) = 14.62/6.77$, $p < 0.05$). cTBS presumably decreased activity in this region further, which is inconsistent with a simple mapping between univariate activity and behavior (i.e. further decreasing activity may have predicted a behavioral improvement). Moreover, the VLPFC showed increased activation during *Contextual Control* (previous/present sample: $F(1,23) = 43.65/126.82$, $p < 10^{-5}$), and verbal processing (previous/present sample: $F(1,23) = 25.8/16.05$, $p < 0.001$), but not their interaction (previous/present sample: $F(1,23) = 3.29/3.32$, $p > 0.05$). Yet, cTBS to this region specifically impaired the interaction among these processes. That behavioral impairments following cTBS do not necessarily track simple regional increases or decreases in activity underscores the need to inform predictions of neural perturbation by models of neural dynamics.

Causal methods provide strong tests of hierarchical relationships. The logic of this approach can be understood by visualizing hierarchy as a pyramid — perturbation of the lowest levels of the hierarchy affect all levels above, whereas perturbation of the top level of the hierarchy does not disrupt the lower levels. The underlying neural mechanism of this hierarchical organization can be understood by the flow of information within hierarchical networks. Lesion data have been used to examine the organization of the LPFC and its hierarchical dependencies. *Badre et al. (2009)* found that

damage to caudal LPFC impaired the ability to base responses on color-feature associations (Feature Task, lower-level deficit). This function is similar to *Feature Control* examined here, with both studies demonstrating a critical role of caudal LPFC. By contrast, *Badre et al. (2009)* found that damage to mid LPFC impaired the ability to base responses on color-feature dimension associations (Dimension Task, higher-level deficit). This function has semblance to *Contextual Control* studied here in that both cases require the use of a higher-order context to determine which feature–feature associations govern the appropriate response. Since Badre et al. did not examine different stimulus domains, it is impossible to tell whether the observed deficit was also stimulus-selective, as was observed here. Critically, *Badre et al. (2009)* demonstrated both hierarchical dependence and selectivity in the reported deficits. Damage to a lower-order controller led to non-parametric deficits at higher levels, but not vice versa (e.g. damage to caudal LPFC impaired the Dimension Task *uniformly* across demands, but damage to the mid LPFC did not impair the Feature Task at all). Furthermore, parametrically varying demands on a given control function revealed parametrically varying deficits when the region involved in that control function was damaged (e.g. damage to caudal LPFC caused *parametrically varying* deficits for the Feature Task, but deficits in the Dimension Task were uniform). This pattern of results is similar to those observed here – cTBS to caudal LPFC was selective at the level of function (i.e. impaired spatial *Feature Control*), and general at higher levels (i.e. was present during, but did not interact with other control demands). Similarly, cTBS to mid LPFC was selective to the level of function (i.e. impaired verbal *Contextual Control*), but did not influence lower levels (i.e. did not impair verbal *Feature Control* more generally). Hence, the pattern of deficits are consistent across these studies, suggesting that similar neural dynamics are engaged by the studied tasks.

*Azuar et al. (2014)* also reported hierarchical dependencies along the rostral-caudal axis using tasks spanning three levels of control: color-response mapping (Sensory Control, low), color-task-response mapping (Contextual Control, mid), episode-color-task-response mapping (Episodic Control, high). In correspondence to the results here, damage to caudal LPFC impaired Sensory Control, damage to mid LPFC impaired Contextual Control, and damage to rostral LPFC impaired Episodic Control. Furthermore, *Azuar et al. (2014)* reported hierarchical dependencies among these deficits such that higher-level deficits followed lower-level deficits, but not vice versa. Unlike *Badre et al. (2009)*, cognitive control demands at each level were not parametrically varied, nor were factorial contrasts used to isolate different control processes. This is potentially problematic in that the tasks used to examine higher levels of control (e.g. Episodic Control) naturally contained the lower levels of control (e.g. Sensory Control). Hence, a patient with an impairment in Sensory Control could show poor performance in the Episodic Control task due to the Sensory Control impairment impacting Episodic Control (hierarchical deficit), or simply due the Sensory Control aspects of the Episodic Control task (non-hierarchical deficit). Hence, it is difficult to establish correspondences between the reported patterns of deficits and those described here. Nevertheless, the pattern of deficits observed by *Azuar et al. (2014)* are consistent with a rostral-to-caudal cascade of activity whereby progressively rostral areas influence caudal areas as the level of cognitive control demands increases (*Koechlin et al., 2003*; *Kouneiher et al., 2009*). A top-down cascade is ostensibly at odds with the dynamics reported here, which demonstrate a convergence of influences towards the mid LPFC. However, these data may be reconciled by considering the caudal LPFC in more detail. In *Koechlin et al. (2003)*, the caudal LPFC is assigned an output role, establishing the appropriate motor response. By contrast, here, the caudal LPFC is assigned an input role, establishing stimulus features for other cognitive control processes. It is likely that caudal LPFC is comprised of both input- and output-related regions, with roles depending upon contrasts used to isolate the caudal LPFC. Furthermore, neural dynamics probably depend upon the role of the investigated region (i.e. output - receives modulations from PFC, input - sends modulations to PFC). Future work investigating both input- and output-related regions simultaneously would help to better illuminate the neural dynamics of cognitive control.

TMS studies have been less convincing with regard to demonstrating hierarchical dependencies among PFC areas. *Bahlmann et al. (2015b)* applied TMS to rostral and caudal LPFC, previously identified with Episodic and Response Control, respectively (*Bahlmann et al., 2015*). TMS to rostral LPFC selectively impaired Episodic Control, but TMS to caudal LPFC produced no measurable impairments. The former result is in agreement with *Azuar et al. (2014)*, whereas the latter result is seemingly at odds with the results of *Azuar et al. (2014)* wherein lesions to caudal LPFC impaired performance on all tasks. A notable difference among these studies is that the caudal LPFC area

identified by *Azuar et al. (2014)* was localized dorsally, at or near the SFS, whereas the caudal LPFC area targeted in *Bahlmann et al. (2015b)* was localized ventrally, at or near the IFJ. If Episodic Control corresponds to *Temporal Control* studied here, an effect of SFS but not IFJ TMS on Episodic Control would be consistent with our data and revised model. Finally, *Rahnev et al. (2016)* applied cTBS to caudal, mid, and rostral LPFC during a perceptual decision-making task. Consistent to the results reported here, cTBS to caudal LPFC impaired spatial attention. cTBS to mid LPFC (correspondingly most closely with MFG here) impaired trading off between fast and accurate responding, a domain-general form of *Contextual Control* that our model would predict to depend upon MFG. Finally, cTBS to rostral LPFC improved meta-cognitive accuracy. This latter result has no clear correspondence in the present task. Regardless, independent effects were observed following cTBS to each PFC area, which showed no discernable hierarchical pattern. However, the lack of hierarchical dependence in this case is probably due to the task wherein each demand could be performed largely independently. We would conjecture that process interdependence is key for examining hierarchical interactions. In such cases, predictions of the effects of focal disruption would be facilitated by models of neural dynamics.

Other TMS studies have examined behaviors relevant to those studied here without directly linking them to models of neural dynamics and hierarchy. *Mottaghy et al. (2002)* found that TMS to dorsal caudal LPFC (at or near SFS) selectively impaired spatial, but not verbal working memory, whereas TMS to ventral caudal LPFC (at or near IFJ) selectively impaired verbal but not spatial working memory. Such results are consistent with our model wherein SFS maintains spatial stimulus features, whereas IFJ maintains verbal stimulus features, which would support spatial and verbal working memory, respectively. Other studies targeting areas at or near the area we have referred to as 'VLPFC' have reported TMS-related reductions in semantic association (*Cattaneo et al., 2009*; *Hoffman et al., 2010*; *Whitney et al., 2011*), long-term memory (*Lee et al., 2013*; *Blumenfeld et al., 2014*), and binding multiple features in working memory (*Morgan et al., 2013*). It is unclear whether a common function underlies all of these cases, but each involves associating identity-based (i.e. ventral stream) representations in a contextual manner. Finally, TMS to the FPl impacts the integration of temporally distal representations into ongoing cognition (*De Pisapia et al., 2012*), and ordering tasks in a sequence in the absence of overt cues (*Desrochers et al., 2015*). These cases are reminiscent of the *Temporal Control* studied here that relies on FPl. Collectively, these data show broad consistencies in effects. Our study further embeds these effects within a model of neural dynamics.

Our original report challenged prevailing accounts positing that the rostral-most LPFC (i.e. FPl) is the apex of the LPFC hierarchy. By definition, a higher place in a hierarchy entails greater influence over that which is lower and vice versa (*Badre and D'Esposito, 2009*). Previously, we found that the mid-rostral PFC (i.e. MFG) showed the presumed signature of a hierarchical apex – greater outward than inward influence in fixed connectivity. This connectivity pattern was also observed in the present dataset, and in the revised model in both datasets. Furthermore, we previously observed that modulations of connectivity dynamics indicated that, during cognitive control, activity converges towards mid LPFC. This result was also observed in the revised model, although some of the pathways along which information flows through the network have been altered. Nevertheless, it is clear that activity diverges (fixed connectivity) and converges (connectivity modulations) to/from the middle of the LPFC in the modeled neural dynamics. Collectively, these data corroborate the hypothesis that we posited previously, that the mid LPFC is a critical nexus for cognitive control. Other work has demonstrated that areas of functional convergence, referred to as dynamic hubs, are particularly important for behavior (*Osada et al., 2015*). While the described work has described hubs within the PFC, hubs that connect networks more globally throughout the brain are also particularly important with damage to such regions producing profound global brain reorganization (*Gratton et al., 2012*) and behavioral impairments (*Warren et al., 2014*). Collectively, such data suggest that importance for brain function and behavior depend critically on where information converges, rather than where it arises.

While the significant effects observed in the present study were by-and-large predicted by a model of LPFC dynamics, the effects were generally limited by task accuracies that were near ceiling (average performance across the task ~95%). As error trials are typically considered separately in fMRI data, we included extensive practice to ensure that participants could perform the task with maximizing power for the fMRI data in mind. Nevertheless, we did not anticipate that many of the

behavioral modulations resulting from cTBS would be borne out in errors. A more suitable procedure may have been to titrate the task to individual performance levels, potentially by requiring adaptively speeded responses to keep performance from ceiling. Given these ceiling effects, it is difficult to interpret null effects in the cTBS data, even though null effects are also important for validating the model. It is also difficult to model the behavioral data (e.g. drift-diffusion modeling) to make inferences regarding the effects of cTBS on different aspects of the decision process (e.g. drift rate, bias). Future explorations using these methods would do well to keep these considerations in mind.

Another limitation of the present work is the temporally tonic nature of cTBS neural modulations. This method precludes inference of precisely when a given region is necessary for behavior. A more time-resolved approach using single pulses or short bursts of TMS (e.g. *Desrochers et al., 2015*) could further elucidate the temporal necessity of each region to cognitive control. Hierarchical dependencies could potentially be revealed by relative orderings of necessity for behavior (e.g. a parent node would be necessary earlier in the trial epoch than a child node). Such methods would be a useful avenue for future investigation.

## Materials and methods

Many of the materials and methods used in this work are identical to those described in our previous report (*Nee and D'Esposito, 2016*). We present abbreviated details here, highlighting differences as well as the cTBS procedures.

### Participants

We report results from 24 (15 female) right-handed native English speakers (mean age 20.5 years, range 18–27). Informed consent was obtained in accordance with the Committee for Protection of Human Subjects at the University of California, Berkeley.

The targeted number of participants was based upon previous work. A sample size of 24 participants was acquired to match to our previous study, which was well-powered to examine fMRI effects. We also considered the efficacy of cTBS. A previous study involving a subset of the authors used a similar cTBS design with a smaller sample size of 17 (*Rahnev et al., 2016*), suggesting that our target sample would probably be adequate. In addition, a sample size of 24 offered the ability to counter-balance the order of cTBS stimulation targets perfectly. All 24 participants are included for the fMRI analyses. In one participant, the control site (S1) was mistargeted, resulting in stimulation of an area that was active for the task (superior parietal lobule). The cTBS data from this participant are excluded. Another participant appeared to misunderstand the instructions for the *Restart* condition, demonstrating below-chance accuracy on return trials. This participant was excluded for all cTBS analyses that included cognitive control demands, but was included in the analysis of *Stimulus Domain* with trials from the *Restart* condition excluded. Inclusion of these data was deemed appropriate over outright exclusion, given that the analysis of *Stimulus Domain* was orthogonal to the cognitive control demands, and given the costs in data acquisition. These exclusions resulted in 23 participants for cTBS analyses focused solely on *Stimulus Domain*, and 22 participants for the remainder of the cTBS analyses.

Three participants performed the fMRI session but were excluded from cTBS sessions. For two participants, functional mapping revealed right-lateralized language processing despite self-reported right-handedness. Another participant was excluded due to an incidental finding. Three participants did not complete all of the cTBS sessions and were therefore excluded from analyses. Two of these participants experienced transient discomfort during stimulation of the rostral LPFC target, which mandated the abortion of that session and subsequent sessions. One participant withdrew from the study after the first cTBS session. These data were excluded from all analyses. In addition, due to a technical error, the final two runs of data were not collected for one session for one participant.

### Procedure

The task design was a factorial 2 × 2 × 2, with factors of *Stimulus Domain* (verbal, spatial), *Contextual Control* (high, low), and *Temporal Control* (high, low). The procedure was nearly identical to that described in our previous report and details can be found there (*Nee and D'Esposito, 2016*). Here we report differences in the procedure.

Whereas previously, participants performed two fMRI sessions, only a single fMRI session was performed here. The number of trials of the basic task at the beginning of each block was reduced from 2–5 trials to 2–3 trials, which helped to improve trial yield for trials-of-interest in the cTBS sessions. Otherwise, the fMRI session was identical to the previously described procedure.

Timing details were mistakenly omitted in our previous report. For fMRI sessions, stimuli were presented for 500 ms followed by a variable inter-trial interval (ITI) of 2600–3400 ms. Feedback indicating the number of correct trials in the block out of the total number of trials in the block was presented for 500 ms at the end of the block. Blocks were spaced by a variable 2600–3400 ms interval. Self-paced breaks were administered in-between runs. For cTBS sessions, the ITI was reduced to 1500–1900 ms, and the interval between blocks was reduced to 1100–1900 ms in order to improve trial yield. Breaks between runs were fixed at 30 s. With these timing reductions, 8 runs were administered, resulting in 16 blocks for each cell of the design. It has been reported that reduced cortical excitability following cTBS begins approximately 5 min post-stimulation and lasts until approximately 60 min post-stimulation (*Huang et al., 2005*). Our timing procedures closely matched this window. A practice run of the task that was half the duration of the experimental runs immediately followed cTBS. As a result, experimental runs began approximately 5 min post-stimulation. A preliminary cursory inspection of run effects (indexed by run within session) revealed no appreciable patterns, so effects of run were not further explored.

Participants performed six sessions scheduled as follows: participants first completed a behavioral session wherein they were introduced to the task and given extensive practice as previously described. Assays of higher-level cognitive ability were also collected as previously described. The fMRI session followed on a separate day, within 10 days of the behavioral session (one participant was scanned 17 days after the behavioral session due to a scheduling issue). The first cTBS session followed within 10 days of the fMRI session (12 days for one participant with a scheduling issue). All cTBS sessions were spaced by one week, with each session beginning at the same time of day for a given individual. The order of stimulation targets was fully counterbalanced and randomized across participants.

## fMRI analysis

Analysis of the fMRI data used methods that were virtually identical to those described previously (*Nee and D'Esposito, 2016*). Regions-of-interest (ROI) analyses were centered on the peak coordinates of our previous report. The one exception to this was for the caudal middle frontal gyrus (cMFG), which was centered on peak activation for *Contextual Control* in the present sample (−26 14 52) to accommodate individual variability in slice prescription that left out the previous peak (−34 10 60) in some participants. ROIs were used to explore abstraction effects both by analysis of *Stimulus Domain* and by correlations with behavior, as previously described.

## Transcranial magnetic stimulation procedures

Transcranial magnetic stimulation (TMS) was delivered using a MagStim Super Rapid2 stimulator equipped with a figure-eight double air film coil with a 70 mm diameter. Electromyography (EMG) was recorded using electrodes placed on the first dorsal interosseus (FDI) muscle on the right-hand using a Delsys Bagnoli system (Delsys Inc.). Individual active motor threshold (AMT) was determined immediately before each session of cTBS. First, a 'hunting procedure' was used to determine the scalp location in the left hemisphere producing the maximal contralateral hand twitch at the minimal stimulation intensity. Next, the participant maintained voluntary contraction of the FDI muscle at ~20% of maximum contraction. AMT was determined as the minimal stimulation intensity needed to produce a motor-evoked potential in 5 out of 10 pulses. Across sessions and participants, the average AMT was 48% of the stimulator output.

cTBS was delivered in a standard manner as described by *Huang et al. (2005)*. Bursts of three pulses at 50 Hz were delivered every 5 Hz for a total of 600 pulses over 40 s. Stimulation intensity was delivered at 80% of the individual's AMT. During all TMS procedures, participants were continually monitored and verbally queried for signs of pain, dizziness, or other adverse effects. Just prior to the start of cTBS, a single test pulse was delivered to the target site at the same intensity of cTBS. The test pulse was used to help the participant determine whether stimulation of the target would be painful. As reported above, two participants found stimulation of rostral LPFC painful during this

test and were excluded from further participation. No adverse effects were reported in the sample used for analysis.

## cTBS targets

cTBS targets were defined on the basis of individual activation maxima (LPFC) or anatomy (S1). We restricted targets to the left hemisphere. LPFC targets were chosen to dissociably stimulate caudal, mid, and rostral LPFC. As reported previously, both dorsal and ventral peaks exist along the rostral-caudal axis of the LPFC. The choice of which target to stimulate for each rostral/caudal sector was motivated by the desire to maximize the distance between targets. Furthermore, given that the ventral caudal LPFC area (i.e inferior frontal junction; IFJ) was distanced from the scalp surface (mean MNI peak −38 6 26 previously), the dorsal caudal LPFC was selected as a more suitable target. This led to the targeting of the ventral mid LPFC and FPl rostrally.

The caudal LPFC target was chosen as the maximal activation for the *Stimulus Domain* contrast, whose sign was dictated by the contrast of *Spatial Stimulus Domain > Verbal Stimulus Domain* (mean MNI location: −23.9–0.6 55.6). Previously, it was found that the mid LPFC shows univariate sensitivity to both *Stimulus Domain* and *Contextual Control*, but not to their interaction. As can be observed in the whole-brain univariate maps (e.g. *Figure 2*), the *Contextual Control* contrast elicits activation along much of the rostral-caudal axis of the LPFC, potentially rendering identification of the appropriate stimulation target difficult on an individual basis. We thus chose to use the *Stimulus Domain* contrast to define the mid LPFC target as this contrast produced more confined activations that were anticipated to reduce variability in target location across participants. In this case, areas were defined as those showing greater activity for the *Verbal Stimulus Domain > Spatial Stimulus Domain*. The ventral mid LPFC area previously described extended from the inferior frontal sulcus (IFS) into the inferior frontal gyrus ventrally and into the middle frontal gyrus dorsally. To maintain consistency, we chose targets restricted to the inferior frontal gyrus. As a result, we now describe the ventral mid LPFC area as the ventrolateral prefrontal cortex (VLPFC) as this term more accurately describes the targeted site. For the first six participants, the maximal activation on the lateral surface of the inferior frontal gyrus was chosen as the stimulation target. At this point, it was determined that this choice of target definition was producing targets that were consistently caudal to the ventral mid LPFC area previously described (mean MNI y-coordinate of the first six participants = 14.0 compared to 20 previously). The procedure was therefore adjusted to target the rostral-most local maxima within pars triangularis and pars opercularis. This resulted in targets that were more clearly situated in mid LPFC (mean MNI location: −51.3 17.5 17.5).

The rostral LPFC target was chosen to be in the vicinity of the FPl previously described (mean MNI location: −34.1 56.9 0.7). In most cases, the rostral LPFC target was identified as the maximal activation for the conjunction of *Temporal Control* and *Contextual Control*. However, if the activation peak was deemed too close to the orbits to stimulate without pain, a more dorsal local maxima was used. In five cases, a target could not be identified through the conjunction contrast. In these cases, either the main effect of *Temporal Control*, main effect of *Contextual Control*, or the simple contrast of Dual > Control was used with the main-effect contrasts taking precedence. This procedure resulted in a target that, on average, was equivalently engaged during *Temporal Control* and *Contextual Control* (average t-statistic for *Temporal Control* – 4.15, average t-statistic for *Contextual Control* – 4.38). Finally, the control site (S1) was chosen as the dorsal-most portion of the post-central gyrus defined anatomically. cTBS targets were guided by a frameless stereotactic localization system (Brainsight; Rogue Research, Inc.).

## cTBS analysis

Behavioral modulations following cTBS were analyzed using linear mixed effects modeling in SPSS version 23 and repeated measures ANOVAs. For the analysis of *Stimulus Domain*, all trials were considered in order to maximize power. This was because the demand to select a stimulus feature (either verbal or spatial) was present in all trials of the task with the potential exception of the sub-task phase of *Delay* trials (notably, removing Delay trials from the analyses does not change the results). Other analyses were restricted to trials from the sub-task phase wherein the other cognitive control demands were manipulated (e.g. *Temporal Control* was not present before and after the sub-task phase). Reaction times (RTs) were computed from correct trials only. RTs greater than 2.5

standard deviations of the condition mean were discarded as outliers, as were trials with RTs less than 200 ms or greater than 2000 ms. ~ 3.5% of the trials were removed as outliers. Including these outliers in the analyses weakens, but does not fundamentally alter, the pattern of results. For analyses on accuracy/error-rate, condition-wise measurements of % correct were arc-sine transformed to normalize the distributions. Accuracies at ceiling (i.e. 100% correct) were corrected prior to transformation with the formula $(n–0.5)/n$, where $n$ is the number of trials in the condition. Statistics were performed on these transformed data. To present error-rates, transformed accuracies were subtracted from the corrected transformed ceiling. Error-rates are presented rather than accuracies to facilitate interpretation alongside RTs, such that greater numbers consistently reflect worse performance.

All linear mixed effects models include a factor of *Time* to account for practice-related changes over sessions. These models also included random effects terms of *Subject*, and *Time* nested within *Subject* (i.e. *Time | Subject*). No assumptions were made of the covariance structure of the random effects terms, which was modeled using restricted maximum-likelihood estimation. Because of the rank of the data, models could only include either *Time | Subject* or *Target | Subject*. Although both effects may be present, the inclusion of *Time | Subject* offered superior model log likelihoods of the covariance structure than did the inclusion of *Target | Subject*, so *Time | Subject* was modeled. In order to simplify the covariance structure, the modeled data were contrasts of interests: *Stimulus Domain* = spatial task – verbal task; *Stimulus Domain x Contextual Control* = [(verbal Dual + verbal Restart) – (verbal Delay + verbal Control)] – [(spatial Dual + spatial Restart) – (spatial Delay + spatial Control)]; *Contextual Control* = (Dual + Restart) – (Delay + Control); *Temporal Control* = (Dual + Delay) – (Restart +Control). Given the assumptions and choices inherent in linear mixed effects modeling, we also present standard full factorial repeated measure ANOVAs (which exclude the factor of *Time*) to provide statistical analyses with fewer assumptions (*Figure 3—figure supplement 1*). Upon finding significant linear mixed model effects of *Target*, t-tests directly comparing Targets were performed after controlling for the effects of *Time*. These analyses were performed on the residuals after regressing out both *Time* and *Time | Subject*.

To test for contrast x Target interactions, the contrasts of the main effect of *Stimulus Domain* in ER, *Stimulus Domain* x *Contextual Control* in RT, and *Temporal Control* in ER were computed (after controlling for *Time* as above) and z-scored across individuals. To make the signs consistent, contrasts containing *Stimulus Domain* were computed such that positive values indicated worse performance on the verbal relative to the spatial task. These were entered into a $3 \times 4$ repeated measures ANOVA. A follow-up $4 \times 4$ ANOVA added the contrast of *Contextual Control* in ER.

## Dynamic causal modeling

DCM was originally performed as previously described (*Nee and D'Esposito, 2016*). The presence/absence of connectivity pathways and modulations was matched to our previous report and the model was estimated in the present sample. Inferences on the parameter estimates were performed in a manner identical to that described previously.

Given the unanticipated effect of cTBS to caudal LPFC on *Temporal Control*, additional DCM analyses were performed on the present sample. First, the original model was compared to two other families of models: 1) fixed connectivity between the SFS and FPl along with modulations (either feedforward, feedback, or bi-directional) as a function of *Temporal Control*, and 2) fixed connectivity between the SFS and MFG along with modulations (either feedforward, feedback, or bi-directional) as a function of *Temporal Control*. Bayesian model selection was first performed family-wise (*Penny et al., 2010*). In this comparison, families including connectivity between SFS and MFG were overwhelmingly favored (model exceedance probability = 0.9992). Given this result, all subsequent model comparisons included fixed connectivity between the SFS and MFG, with modulations as a function of *Temporal Control* (feedforward, feedback, bi-directional, or absent) varying across models. We used an iterative procedure as previously described to explore the model space while maintaining computational feasibility (*Nee and D'Esposito, 2016*). Here, the model space was identical to that described before with the addition of the possibility of modulations between the SFS and MFG as a function of *Temporal Control*. In this case, the iterative procedure did not converge, instead hitting an infinite loop. In total, 913 models were explored. Simultaneous comparison of these models using Bayesian model selection (*Stephan et al., 2009*) did not identify a single best model in terms of an exceptional model exceedance probability. A scree plot of the models ordered

by their exceedance probabilities identified potential step-wise differences for the top 18 and top 27 models. It is possible that the failure to identify a clearly superior model was due to the large number of models explored that overlapped in many features. Hence, Bayesian model selection was performed on a reduced model space taking the top 27 models. Once again, no single model was clearly favored, but the top 19 models (all model exceedance probability ~0.04) were ordered similarly to before and were once again consistently favored over the other models (model exceedance probabilities ~0.02). Nevertheless, these model exceedance probabilities were no different than uniform (1/27 = 0.037).

In order to better adjudicate between the models of LPFC dynamics, the best 19 models identified above were compared within the previous sample. Given that the previous sample contained twice the fMRI data per subject, this offered the ability to produce more stable model parameter estimates with better potential to delineate among the models. The original model was also included for comparison to verify that the addition of SFS to MFG connectivity was prudent, making a total of 20 models for comparison. In this case, far more convincing evidence was found for the top model (model exceedance probability = 0.2526, compare to a uniform of 0.05). Other models with better-than-uniform model exceedance probabilities shared the main features that were found to be significant in the data reported in the main text (feedforward modulation from SFS to MFG for *Temporal Control*, feedback modulation from FPl to MFG and MFG to cMFG/VLPFC for *Contextual Control*), bolstering the credibility of these effects. Parameter estimates for the best model were examined for the previous dataset and the present dataset, and compared for consistency. Noting good agreement among the parameter estimates (*Figure 4—figure supplement 1*), the two datasets were combined for subsequent inference. Random effects inference on the parameters of the winning model then proceeded in a manner identical to our previous report.

To summarize:

i) The winning DCM from *Nee and D'Esposito, 2016* was estimated on the present dataset.

ii) New DCM's were estimated on the present dataset to examine whether the observed effect of cTBS on SFS on *Temporal Control* could be accommodated by a revised model.

iii) The top 19 models from (ii) were difficult to adjudicate among, and were subsequently compared to each other and the winning DCM from *Nee and D'Esposito, 2016* in the previous dataset.

iv) Parameter estimates from the best model in (iii) were examined in both the present and previous datasets and pooled for inference.

## Acknowledgements

This research was supported by National Institute of Neurological Disorders and Stroke Grants F32 NS0802069 (DN) and P01 NS040813 (MD), and by National Institute of Mental Health Grant R01 MH063901 (MD). The authors thank Dobromir Rahnev for helpful advice with the TMS procedure, and Max Wang, Lara Krisst, Grace Lee, Joanne Lin, and Chelsea Harmon for help with data collection.

## Additional information

### Funding

| Funder | Grant reference number | Author |
|---|---|---|
| National Institute of Neurological Disorders and Stroke | F32 NS0802069 | Derek Evan Nee |
| National Institute of Neurological Disorders and Stroke | P01 NS040813 | Mark D'Esposito |
| National Institute of Mental Health | R01 MH063901 | Mark D'Esposito |

The funders had no role in study design, data collection and interpretation, or the decision to submit the work for publication.

## Author contributions
Derek Evan Nee, Conceptualization, Data curation, Software, Formal analysis, Funding acquisition, Investigation, Visualization, Methodology, Writing—original draft, Project administration, Writing—review and editing; Mark D'Esposito, Conceptualization, Supervision, Funding acquisition, Writing—review and editing

## Author ORCIDs
Derek Evan Nee http://orcid.org/0000-0001-7858-6871

## Ethics
Human subjects: Informed consent was obtained in accordance with the Committee for Protection of Human Subjects at the University of California, Berkeley under protocol number 2010-02-781.

## Decision letter and Author response
Decision letter https://doi.org/10.7554/eLife.28040.018
Author response https://doi.org/10.7554/eLife.28040.019

# Additional files

## Supplementary files
• Transparent reporting form
DOI: https://doi.org/10.7554/eLife.28040.014

## Major datasets
The following dataset was generated:

| Author(s) | Year | Dataset title | Dataset URL | Database, license, and accessibility information |
|---|---|---|---|---|
| Derek Evan Nee, Mark D'Esposito | 2017 | Causal Evidence for Lateral Prefrontal Cortex Dynamics Supporting Cognitive Control | https://oneshare.cdlib.org/stash/dataset/doi:10.15146/R3Q37R | Publicly available on Oneshare |

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
