## [Decision Letter]

Thank you for submitting your article "Causal Evidence for Lateral Prefrontal Cortex Dynamics Supporting Cognitive Control" for consideration by *eLife*. Your article has been reviewed by two peer reviewers, and the evaluation has been overseen by a Reviewing Editor and Timothy Behrens as the Senior Editor. The following individual involved in review of your submission has agreed to reveal his identity: Charan Ranganath (Reviewer #2).

The reviewers have discussed the reviews with one another and the Reviewing Editor has drafted this decision to help you prepare a revised submission.

The reviewers both commend the authors for providing a nice replication of your previous fMRI work dissociating LPFC regions with further manipulation of those circuits using stimulation methods and testing the predictions from their dynamic causal modeling results.

However, they agreed that there were several issues that need to be addressed to clarify the strength of the findings and better understand what the implications are for broadening our understanding of PFC function.

1) While you state that the findings in ER were unexpected, there is little explanation for why some effects would be observed in RT and some in ER? Given the emphasis on modeling and model testing, it would be nice to see an attempt at a testable account for this difference among the nodes. Relatedly, please present the full results for both RT and ER in the main figure (instead of in the supplemental).

2) From a statistical perspective, the behavioral effects of TMS, as framed in the paper, are relatively weak. This may be due to the choice of analysis approaches. The 2x2x4 ANOVA approach is nice in the sense that it makes few assumptions, but it is not powerful for detecting the predicted stimulation target-related interactions because of the nonsphericity correction. Please think outside of the box on this issue – all ANOVAs are effectively fitting particular models to the data, but they are just non-optimal models for a study like this. One alternative approach would be to run an omnibus F-test to first show that there is something going on, and then to break it down by fitting a model of the exact pattern of results predicted by the authors' model for each stimulation approach. This model can be tested against reduced or alternative models. That's just a suggestion, however, the point is that the current approach of reporting nonsignificant interactions and then running planned comparisons seems to undermine the whole ANOVA approach in the first place.

3) You nicely show that if the model is updated, better model fits are obtained based on including further potential connectivity and modulators. Is this true for any of the other nodes in the network? Particularly the IFJ? If SFS and IFJ has a similar function but in different modalities, then it might follow that their influence on the network might be similar? This space should be explored. It also raises the question of other nodes having unanticipated connections. While testing all possibilities is clearly not feasible, showing that other potential connections (e.g. FPl to SFS) do or do not produce changes in model fit could help clarify the proposed functioning of the network.

4) Please discuss how the current results should alter our view of PFC function. It was noted by a reviewer that "the authors describe various findings in the literature and conclude that, "Collectively, these results indicate that whether or not hierarchical dependencies exist depends critically upon task-elicited regional interdependencies." It is a concern that this statement *completely nullifies* the significance of the study – if the presence or absence of a hierarchy is completely task dependent. Please reconcile this with the statement you make that the "data provide causal evidence for LPFC dynamics supporting cognitive control".

5) Please address the issue of practice more thoroughly. Does it have any effect on the modeling and relationship to cognitive capacity results – especially given that there were differences? A cursory examination of within-session practice effects was made but not elaborated upon. Given that the participants are participating in many, spaced sessions, the effect of session should be examined both in isolation and as a covariate. There is the potential for effects to interact with practice.

6) Please justify the use of a one-tailed test for the comparison between SFS and S1. Was there an a-priori reason for doing so? If not, this should not be included as a significant result. In terms of the analysis of the cTBS results, please clarify why for all but Stimulus Domain analyses a subset of trials were analyzed. Also, with some of the TMS effects in RT, why remove long RT's? Was there a systematic kind of trial that was removed with this process? Does including those RTs alter the results in any way?

7) Was the entire DCM run on the new sample only or on the combined data from the current sample and the previous *eLife* publication? It looks like some combination of the two, which is odd especially because the initial replication DCM was performed separately on the new dataset (right?). Some clarification is warranted.

---

## [Author Response]

The reviewers both commend the authors for providing a nice replication of your previous fMRI work dissociating LPFC regions with further manipulation of those circuits using stimulation methods and testing the predictions from their dynamic causal modeling results.However, they agreed that there were several issues that need to be addressed to clarify the strength of the findings and better understand what the implications are for broadening our understanding of PFC function.1) While you state that the findings in ER were unexpected, there is little explanation for why some effects would be observed in RT and some in ER? Given the emphasis on modeling and model testing, it would be nice to see an attempt at a testable account for this difference among the nodes. Relatedly, please present the full results for both RT and ER in the main figure (instead of in the supplemental).

Our expectation that effects of TMS would be observed primarily in RT was driven by a simple conception of an accumulator decision process wherein cTBS to a given region would slow the rate of evidence accumulation by way of inhibiting neuronal activity (e.g. the inverse of what has been observed via microstimulation in Hanks et al., 2006, Nat Neurosci). Whether effects that are borne out in ER rather than RT are telling regarding the decision process is unclear to us. RT and ER tend to tradeoff and subjects can strategically shift between the two (e.g. Fitts et al., 1966, JEP), and we have previously demonstrated that such shifts are under PFC control (Rahnev et al., 2016, PNAS). While deeper exploration of the locus of the effect (i.e. ER vs. RT) and potential interactions with speed-accuracy tradeoffs strategies would be interesting, the low number of errors by condition renders it difficult to model the decision process (e.g. drift diffusion modeling). While we have made some attempts to do so, we could not achieve satisfactory model fits. Given that we cannot gain theoretical traction on the matter, we refrain from speculating why some effects are observed in ER and others in RT. Instead, we discuss the limitations that the low ERs place on the interpretation of these data in the Discussion.

“While the significant effects observed in the present study were by-and-large predicted by a model of LPFC dynamics, the effects were generally limited by task accuracies that were near ceiling (average performance across the task ~95%). […] Future explorations using these methods would do well to keep these considerations in mind.”

As recommended, the full RT and ER results are now reported in Figure 3.

2) From a statistical perspective, the behavioral effects of TMS, as framed in the paper, are relatively weak. This may be due to the choice of analysis approaches. The 2x2x4 ANOVA approach is nice in the sense that it makes few assumptions, but it is not powerful for detecting the predicted stimulation target-related interactions because of the nonsphericity correction. Please think outside of the box on this issue – all ANOVAs are effectively fitting particular models to the data, but they are just non-optimal models for a study like this. One alternative approach would be to run an omnibus F-test to first show that there is something going on, and then to break it down by fitting a model of the exact pattern of results predicted by the authors' model for each stimulation approach. This model can be tested against reduced or alternative models. That's just a suggestion, however, the point is that the current approach of reporting nonsignificant interactions and then running planned comparisons seems to undermine the whole ANOVA approach in the first place.

A fair point is made that it is somewhat awkward to state the absence of a significant interaction, but follow up with significant planned contrasts. While we appreciate the sentiment of “thinking outside the box” with regard to examining hypothesized effects in a potentially more sensitive manner, we are also cautious about exercising degrees of freedom in analysis so as to avoid overfitting the data. Nevertheless, we believe that a middle ground has been achieved:

In addressing the issue of practice, we have had to adjust the modeling framework. The resultant models have yielded results that strengthen the previously reported patterns. In particular, we now report the results of linear mixed effects models that add time as a covariate in order to account for practice effects over cTBS sessions. As the reviewers keyed in on, many of the cognitive control effects varied with time and accounting for this variance reduced the error of the contrasts of interest. In order to simplify modeling the co-variance structure of the data, the linear mixed effects models are estimated on contrasts of the behavior effects, thereby providing a more streamlined and targeted approach, as recommended. In the present manuscript, significant effects of Target are now observed in either ER or RT for all of the hypothesized interactions. In addition to amending the Results with the aforementioned statistical tests, Figure 3 has been revised to depict the contrasts of interests after controlling for effects of time.

As the reviewers note, ANOVAs are nice in that they have few assumptions and the ANOVAs reported previously were the originally planned analyses. For the sake of transparency and reproducibility, the revised manuscript retains all of these analyses in Figure 3—figure supplement 1 (with slight modification to account for an error detailed in our third response to comment #6). We hope that this presentation strikes a balance between hypothesis driven analysis that controls for confounding factors, and assumption-free analysis.

See Results subsection “Behavioral Results and Effects of cTBS”; Figure 3.

3) You nicely show that if the model is updated, better model fits are obtained based on including further potential connectivity and modulators. Is this true for any of the other nodes in the network? Particularly the IFJ? If SFS and IFJ has a similar function but in different modalities, then it might follow that their influence on the network might be similar? This space should be explored. It also raises the question of other nodes having unanticipated connections. While testing all possibilities is clearly not feasible, showing that other potential connections (e.g. FPl to SFS) do or do not produce changes in model fit could help clarify the proposed functioning of the network.

The reviewers raise a good point about specificity. If adding a bidirectional Temporal Control modulation between SFS and MFG improves the model, it is possible that a similar addition from other nodes (e.g. IFJ to SFS, FPl to SFS) might do the same. In the least, a similar modulation from IFJ would give the model symmetry. To explore this matter, we estimated 8 new models in the current (TMS) sample, and replicated the same analysis in the previous (fMRI only) sample. These models added bidirectional Temporal Control modulations between MFG and each of IFJ, VLPFC, cMFG, or FPl. These modulations were added in addition to, or in place of the bidirectional modulation between SFS and MFG previously described. In both the TMS sample and the fMRI only sample, adding these modulations did not improve the model fit as evidenced by the previously reported model having a clearly superior model exceedance probability in the 9 model comparison (TMS sample exceedance probability = 0.5192, fMRI only sample exceedance probability = 0.8076, compare to a uniform of 0.1111). While exploring the full space of unanticipated connections is computationally infeasible, this result is consistent with specificity with regard to the reported SFS to MFG connection.

From the Results:

“While the analyses above indicate that including a bidirectional modulation between SFS and MFG during *Temporal Control* provides the best fit of the data among the models tested, it remains possible that there may yet be other unexamined modulations that provide an even better fit. […] Such data indicate that there is specificity to the interaction between SFS and MFG during *Temporal Control* and that simply adding additional modulations does not improve the model fit.”

4) Please discuss how the current results should alter our view of PFC function. It was noted by a reviewer that "the authors describe various findings in the literature and conclude that, "Collectively, these results indicate that whether or not hierarchical dependencies exist depends critically upon task-elicited regional interdependencies." It is a concern that this statement completely nullifies the significance of the study – if the presence or absence of a hierarchy is completely task dependent. Please reconcile this with the statement you make that the "data provide causal evidence for LPFC dynamics supporting cognitive control".

The reviewers raise good points. We had previously attempted to keep the Discussion short, attempting to leave the theoretical heavy-lifting in our previous report and making this piece more about A) a test of the theory, and B) a method that can be used to test these and related theories. However, hierarchical cognitive control is a complex literature that may be done a disservice by cursory treatments.

The Discussion has been revised substantially in order to better elucidate the role of PFC in cognitive control. In particular, we make a stronger effort to denote the correspondences between the present data and the literature, as well as explain the discrepancies. We have also fleshed out the main idea about PFC function that we wished to convey in this domain – that perhaps importance should shift from identifying the “top” to identifying hubs where information coalesces.

A last point that we’d like to respond to: we continue to maintain that the presence or absence of hierarchy is task dependent, and we disagree with the statement that this position nullifies the significance of the study. It is our position that the PFC can act hierarchically only insofar as the task demands it. By way of analogy, consider baseball – the manager can exert hierarchical control over the players (e.g. determining who plays, where, and even how). Once the game is in motion, the game often dictates how much hierarchical control the manager should exert. For example, if the team is winning and the players are performing well, the manager does relatively little. In this case, control is left to the players (e.g. the pitcher and catcher determine the pitches, the shortstop orchestrates infield defensive shifts, etc.). However, if the game is tight, the manager is likely to exert more control (e.g. making pitching/batting substitutions, calling for bunts or intentional walks, etc.). It is our position that a similar situation plays out in the PFC. If a task does not call for hierarchical control, it will likely not engage hierarchical dynamics. In this case, even if a region’s role includes exerting hierarchical control, such influences will not be revealed if the task is ill-suited to reveal it. This does not nullify the significance of that region in general, but only for that task. To expect that a PFC region will perform the same function for all tasks is perhaps expecting too much simplicity from an area that is defined by its very adaptability.

See the Discussion.

5) Please address the issue of practice more thoroughly. Does it have any effect on the modeling and relationship to cognitive capacity results – especially given that there were differences? A cursory examination of within-session practice effects was made but not elaborated upon. Given that the participants are participating in many, spaced sessions, the effect of session should be examined both in isolation and as a covariate. There is the potential for effects to interact with practice.

Please see our response to comment #2.

6) Please justify the use of a one-tailed test for the comparison between SFS and S1. Was there an a-priori reason for doing so? If not, this should not be included as a significant result.

As detailed, we predicted that cTBS to SFS would impair spatial processing. The appropriate test for this is to compare spatial processing with a well-matched control (i.e. verbal processing), and SFS to a well-matched control (i.e. S1). Hence, the test that was performed was a planned analysis that was a priori expected to produce the observed result. Thus, we deemed a one-tailed test appropriate. However, after taking effects of time into account (see response 2/3 above), all significant results are significant at two-tailed levels.

In terms of the analysis of the cTBS results, please clarify why for all but Stimulus Domain analyses a subset of trials were analyzed.

For all analyses, all relevant trials were included. For the tests of Stimulus Domain, this includes all trials since on all trials, participants focus attention on either the spatial or object (letter) feature. For tests that examine Contextual Control and/or Temporal Control and/or their interactions, the relevant trials are the sub-task phase during which time those control processes are manipulated. These matters have been clarified in the text.

From the Materials and methods:

“For the analysis of *Stimulus Domain*, all trials were considered in order to maximize power. […] Other analyses were restricted to trials from the sub-task phase wherein the other cognitive control demands were manipulated (e.g. *Temporal Control* was not present before and after the sub-task phase).”

Also, with some of the TMS effects in RT, why remove long RT's? Was there a systematic kind of trial that was removed with this process? Does including those RTs alter the results in any way?

While outlier removal is a common practice in analysis of behavioral data, it is important to ensure that it does not bias the results in some way. To ensure that outlier removal did not systematically remove any one type of trial, outliers were identified condition-wise either by Stimulus Domain (when considering all trials), or by the crossing of Stimulus Domain x Contextual Control x Temporal Control (when considering sub-task trials). This manner of outlier correction ignores the effect of Target. However, it is possible that long RTs may systematically result from one Target or another. This did not appear to be the case as outliers were evenly distributed across Targets (all trials: FPl – 3.6%, VLPFC – 3.6%, SFS – 3.5%, S1 – 3.3%; sub-task trials: FPl – 3.5%, VLPFC – 4.0%, SFS – 3.5%, S1 – 3.8%). In 3 of the 4 planned comparisons, inclusion of these trials tends to reduce interactions with Target, be they significant or non-significant [Target x Stimulus Domain (all trials): p = 0.269 (Corrected RT) -> 0.479 (Uncorrected RT); Target x Stimulus Domain x Contextual Control (sub-task trials): p = 0.002 (Corrected RT) -> 0.081 (Uncorrected RT); Target x Temporal Control (sub-task trials): p = 0.021 (Corrected RT) -> 0.061 (Uncorrected RT); Target x Contextual Control (sub-task trials): p = 0.733 (Corrected RT) -> 0.255 (Uncorrected RT)]. This is to be expected if these outlier RTs add noise to the analysis.

One note: in doing this analysis, it was discovered that error trials were not removed from the RT distributions for factorial analysis of the TMS data due to a coding mistake. Error trials were correctly removed for the fMRI data and for the computation of outlier RTs (i.e. outlier RTs were calculated on correct trials only). Given the small number of error trials, this mistake had a minimal impact on the RT analyses. The manuscript now correctly reports data drawn from correct RT distributions, which have also been cleaned of outliers. This detail accounts for minor discrepancies between the original submission and the data reported in Figure 3 and Figure 3—figure supplement 1.

7) Was the entire DCM run on the new sample only or on the combined data from the current sample and the previous eLife publication? It looks like some combination of the two, which is odd especially because the initial replication DCM was performed separately on the new dataset (right?). Some clarification is warranted.

We regret that these details were not clear. The manuscript has been amended to clarify. To summarize:

i) We estimated the winning DCM from Nee and D’Esposito, 2016 on the present sample as a way of replication.

ii) We estimated new DCM’s on the current sample to examine whether the effect of cTBS of SFS affecting Temporal Control could be accommodated by a revised DCM. The initial intention was to arrive at a best model in the current sample, and then validate this choice in the previous sample by comparing the best *n* models, where *n* would be dictated by a fall-off in model exceedance probabilities.

iii) However, step ii did not fully converge and resulted in 19 (of 913 estimated) equally plausible DCM’s. To adjudicate among these models, we returned to the previous sample which had twice the data, and could potentially offer better power to determine the best model.

iv) The 19 DCM’s mentioned in step iii were compared in the previous sample. The winning DCM from Nee and D’Esposito, 2016 was also included in this comparison. The comparison resulted in a clear winning DCM.

v) Parameter estimates for the winning DCM from step iv is depicted with the samples combined (i.e. averaging across all participants) in Figure 4, and separately for each sample in Figure 4—figure supplement 1.

From the Materials and methods:

“To summarize:

i) The winning DCM from Nee and D’Esposito, 2016 was estimated on the present dataset. […] iv) Parameter estimates from the best model in iii. Were examined in both the present and previous datasets and pooled for inference.”